# Differentiable Learning of Submodular Models

**Josip Djolonga**
Department of Computer Science
ETH Zurich
josipd@inf.ethz.ch

**Andreas Krause**
Department of Computer Science
ETH Zurich
krausea@ethz.ch

## Abstract

Can we incorporate discrete optimization algorithms within modern machine learning models? For example, is it possible to incorporate in deep architectures a layer whose output is the minimal cut of a parametrized graph? Given that these models are trained end-to-end by leveraging gradient information, the introduction of such layers seems very challenging due to their non-continuous output. In this paper we focus on the problem of submodular minimization, for which we show that such layers are indeed possible. The key idea is that we can continuously relax the output without sacrificing guarantees. We provide an easily computable approximation to the Jacobian complemented with a complete theoretical analysis. Finally, these contributions let us experimentally learn probabilistic log-supermodular models via a bi-level variational inference formulation.

## 1 Introduction

Discrete optimization problems are ubiquitous in machine learning. While the majority of them are provably hard, a commonly exploitable trait that renders some of them tractable is that of submodularity [1, 2]. In addition to capturing many useful phenomena, submodular functions can be minimized in polynomial time and also enjoy a powerful connection to convex optimization [3]. Both of these properties have been used to great effect in both computer vision and machine learning, to e.g. compute the MAP configuration in undirected graphical models with long-reaching interactions [4] and higher-order factors [5], clustering [6], to perform variational inference in log-supermodular models [7, 8], or to design norms useful for structured sparsity problems [9, 10].

Despite all the benefits of submodular functions, the question of how to learn them in a practical manner remains open. Moreover, if we want to open the toolbox of submodular optimization to modern practitioners, an intriguing question is how to to use them in conjunction with deep learning networks. For instance, we need to develop mechanisms that would enable them to be trained together in a fully end-to-end fashion. As a concrete example from the computer vision domain, consider the problem of image segmentation. Namely, we are given as input an RGB representation $\mathbf{x} \in \mathbb{R}^{n \times 3}$ of an image captured by say a dashboard camera, and the goal is to identify the set of pixels $A \subseteq \{1, 2, \ldots, n\}$ that are occupied by pedestrians. While we could train a network $\boldsymbol{\theta} \colon \mathbf{x} \to \mathbf{v} \in \mathbb{R}^n$ to output per-pixel scores, it would be helpful, especially in domains with limited data, to bias the predictions by encoding some prior beliefs about the expected output. For example, we might prefer segmentations that are spatially consistent. One common approach to encourage such configurations is to first define a graph over the image $\mathcal{G} = (V, E)$ by connecting neighbouring pixels, specify weights $\mathbf{w}$ over the edges, and then solve the following graph-cut problem

$$A^*(\mathbf{w}, \mathbf{v}) = \underset{A \subseteq V}{\arg\min}\, F(A) = \underset{A \subseteq V}{\arg\min} \sum_{\{i,j\} \in E} w_{i,j} \underbrace{[\![A \cap \{i,j\} = 1]\!]}_{\text{1 iff the predictions disagree}} + \sum_{i \in A} \underbrace{v_i}_{\text{pixel score}} . \quad (1)$$

While this can be easily seen as a module computing the best configuration as a function of the edge weights and per-pixel scores, incorporating it as a layer in a deep network seems at a first glance to

be a futile task. Even though the output is easily computable, it will be discontinuous and have no Jacobian, which is necessary for backpropagation. However, as the above problem is an instance of submodular minimization, we can leverage its relationship to convexity and relax it to

$$\mathbf{y}^*(\mathbf{w}, \mathbf{v}) = \arg\min_{\mathbf{y}\in\mathbb{R}^n} f(\mathbf{y}) + \frac{1}{2}\|\mathbf{y}\|^2 = \arg\min_{\mathbf{y}\in\mathbb{R}^n} \sum_{\{i,j\}\in E} w_{i,j}|y_i - y_j| + \mathbf{v}^T\mathbf{y} + \frac{1}{2}\|\mathbf{y}\|^2. \quad (2)$$

In addition to having a continuous output, this relaxation has a very strong connection with the discrete problem as the discrete optimizer can be obtained by thresholding $\mathbf{y}^*$ as $A^* = \{i \in V \mid y_i^* > 0\}$. Moreover, as explained in §2, for every submodular function $F$ there exists an easily computable convex function $f$ so that this relationship holds. For general submodular functions, the negation of the solution to the relaxed problem (2) is known as the *min-norm point* [11]. In this paper we consider the problem of designing such modules that solve discrete optimization problems by leveraging this continuous formulation. To this end, our key technical contribution is to analyze the sensitivity of the min-norm point as a function of the parametrization of the function $f$. For the specific case above we will show how to compute $\partial\mathbf{y}^*/\partial\mathbf{w}$ and $\partial\mathbf{y}^*/\partial\mathbf{v}$.

Continuing with the segmentation example, we might want to train a conditional model $P(A \mid \mathbf{x})$ that can model the uncertainty in the predictions to be used in downstream decision making. A rich class of models are log-supermodular models, i.e., those of the form $P(A \mid \mathbf{x}) = \exp(-F_{\boldsymbol{\theta}(\mathbf{x})}(A))/\mathcal{Z}(\boldsymbol{\theta})$ for some parametric submodular function $F_{\boldsymbol{\theta}}$. While they can capture very useful interactions, they are very hard to train in the maximum likelihood setting due to the presence of the intractable normalizer $\mathcal{Z}(\boldsymbol{\theta})$. However, Djolonga and Krause [8] have shown that for any such distribution we can find the closest fully factorized distribution $Q(\cdot \mid \mathbf{x})$ minimizing a specific information theoretic divergence $D_\infty$. In other words, we can *exactly* compute $Q(\cdot \mid \mathbf{x}) = \arg\min_{Q\in\mathcal{Q}} D_\infty(P(\cdot \mid \mathbf{x}) \,\|\, Q)$, where $\mathcal{Q}$ is the family of fully factorized distributions. Most importantly, the optimal $Q$ can also be computed from the min-norm point. Thus, a reasonable objective would be to learn a model $\boldsymbol{\theta}(\mathbf{x})$ so that the best *approximate* distribution $Q(\cdot \mid \mathbf{x})$ gives high likelihood to the training data point. This is a complicated bi-level optimization problem (as $Q$ implicitly depends on $\boldsymbol{\theta}$) with an inner variational inference procedure, which we can again train end-to-end using our results. In other words, we can optimize the following algorithm end-to-end with respect to $\boldsymbol{\theta}$.

$$\mathbf{x}_i \xrightarrow{\theta(\mathbf{x})} \boldsymbol{\theta}_i \longrightarrow P = \exp(-F_{\boldsymbol{\theta}_i}(A))/\mathcal{Z}(\boldsymbol{\theta}_i) \longrightarrow Q = \arg\min_{Q\in\mathcal{Q}} D_\infty(P \,\|\, Q) \longrightarrow Q(A_i \mid \mathbf{x}_i). \quad (3)$$

**Related work.**   Sensitivity analysis of the set of optimal solutions has a long history in optimization theory [12]. The problem of argmin-differentiation of the specific case resulting from graph cuts (i.e. eq. (2)) has been considered in the computer vision literature, either by smoothing the objective [13], or by unrolling iterative methods [14]. The idea to train probabilistic models by evaluating them using the marginals produced by an approximate inference algorithm has been studied by Domke [15] for tree-reweighted belief propagation and mean field, and for continuous models by Tappen [16]. These methods either use the implicit function theorem, or unroll iterative optimization algorithms. The benefits of using an inconsistent estimator, which is what we do by optimizing eq. (3), at the benefit of using computationally tractable inference methods has been discussed by Wainwright [17]. Amos and Kolter [18] discuss how to efficiently argmin-differentiate quadratic programs by perturbing the KKT conditions, an idea that goes back to Boot [19]. We make an explicit connection to their work in Theorem 4. In Section 4 we harness the connection between the min-norm problem and isotonic regression, which has been exploited to obtain better duality certificates [2], and by Kumar and Bach [20] to design an active-set algorithm for the min-norm problem. Chakravarti [21] analyzes the sensitivity of the optimal isotonic regression point with respect to perturbations of the input, but does not discuss the directional derivatives of the problem. Recently, Dolhansky and Bilmes [22] have used deep networks to parametrize submodular functions. Discrete optimization is also used in structured prediction [23, 24] for the computation of the loss function, which is closely related to our work if we use discrete optimization only at the last layer. However, in this case we have the advantage in that we allow for arbitrary loss functions to be applied to the solution of the relaxation.

**Contributions.**   We develop a very efficient approximate method (§4) for the computation of the Jacobian of the min-norm problem inspired by our analysis of isotonic regression in §3, where we derive results that might be of independent interest. Even more importantly, from a practical perspective, this Jacobian has a very nice structure and we can multiply with it in linear time. This

means that we can efficiently perform back-propagation if we use these layers in a modern deep architectures. In §5 we show how to compute directional derivatives exactly in polynomial time, and give conditions under which our approximation is correct. This is also an interesting theoretical result as it quantifies the stability of the min-norm point with respect to the model parameters. Lastly, we use our results to learn log-supermodular models in §6.

## 2 Background on Submodular Minimization

Let us introduce the necessary background on submodular functions. They are defined over subsets of some *ground set*, which in the remaining of this paper we will w.l.o.g. assume to be $V = \{1, 2, \ldots, n\}$. Then, a function $F \colon 2^V \to \mathbb{R}$ is said to be submodular iff for all $A, B \subseteq V$ it holds that

$$F(A \cup B) + F(A \cap B) \leq F(A) + F(B). \tag{4}$$

We will furthermore w.l.o.g. assume that $F$ is normalized so that $F(\emptyset) = 0$. A very simple family of submodular functions are modular functions. These, seen as discrete analogues of linear functions, satisfy the above with equality and are given as $F(A) = \sum_{i \in A} m_i$ for some real numbers $m_i$. As common practice in combinatorial optimization, we will treat any vector $\mathbf{m} \in \mathbb{R}^n$ as a modular function $2^V \to \mathbb{R}$ defined as $m(A) = \sum_{i \in A} m_i$. In addition to the graph cuts from the introduction (eq. (1)), another widely used class of functions are concave-of-cardinality functions, i.e. those of the form $F(|A|) = h(|A|)$ for some concave $h \colon \mathbb{R} \to \mathbb{R}$ [5]. From eq. (4) we see that if we want to define a submodular function over a collection $\mathcal{D} \subsetneq 2^V$ it has to be closed under union and intersection. Such collections are known as *lattices*, and two examples that we will use are the simple lattice $2^V$ and the trivial lattice $\{\emptyset, V\}$. In the theory of submodular minimization, a critical object defined by a pair consisting of a submodular function $F$ and a lattice $\mathcal{D} \supseteq \{\emptyset, V\}$ is the base polytope

$$\mathcal{B}(F \mid \mathcal{D}) = \{\mathbf{x} \in \mathbb{R}^n \mid x(A) \leq F(A) \text{ for all } A \in \mathcal{D}\} \cap \{\mathbf{x} \in \mathbb{R}^n \mid x(V) = F(V)\}. \tag{5}$$

We will also use the shorthand $\mathcal{B}(F) = \mathcal{B}(F \mid 2^V)$. Using the result of Edmonds [25], we know how to maximize a linear function over $\mathcal{B}(F)$ in $O(n \log n)$ time with $n$ function evaluations of $F$. Specifically, to compute $\max_{\mathbf{y} \in \mathcal{B}(F)} \mathbf{z}^T \mathbf{y}$, we first choose a permutation $\sigma \colon V \to V$ that sorts $\mathbf{z}$, i.e. so that $z_{\sigma(1)} \geq z_{\sigma(2)} \geq \cdots \geq z_{\sigma(n)}$. Then, a maximizer $\mathbf{f}(\sigma) \in \mathcal{B}(F)$ can be computed as

$$[\mathbf{f}(\sigma)]_{\sigma(i)} = F(\{\sigma(i)\} \mid \{\sigma(1), \ldots, \sigma(i-1)\}), \tag{6}$$

where the marginal gain of $A$ given $B$ is defined as $F(A \mid B) = F(A \cup B) - F(B)$. Hence, we know how to compute the support function $f(\mathbf{z}) = \sup_{\mathbf{y} \in \mathcal{B}(F)} \mathbf{z}^T \mathbf{y}$, which is known as the Lovász extension [3]. First, this function is indeed an extension as $f(\mathbf{1}_A) = F(A)$ for all $A \subseteq V$, where $\mathbf{1}_A \in \{0,1\}^n$ is the indicator vector for the set $A$. Second, it is convex as it is a supremum of linear functions. Finally, and most importantly, it lets us minimize submodular functions in polynomial time with convex optimization because $\min_{A \in 2^V} F(A) = \min_{\mathbf{z} \in [0,1]} f(\mathbf{z})$ and we can efficiently round the optimal continuous point to a discrete optimizer. Another problem, with a smooth objective, which is also explicitly tied to the problem of minimizing $F$ is that of computing the min-norm point, which can be defined in two different ways as

$$\mathbf{y}^* = \arg\min_{\mathbf{y} \in \mathcal{B}(F)} \frac{1}{2}\|\mathbf{y}\|^2, \text{ or equivalently as } -\mathbf{y}^* = \arg\min_{\mathbf{y}} f(\mathbf{y}) + \frac{1}{2}\|\mathbf{y}\|^2, \tag{7}$$

where the equivalence comes from strong Fenchel duality [2]. The connection with submodular minimization comes from the following lemma, which we have already hinted at in the introduction.

**Lemma 1** ([1, Lem. 7.4]). *Define $A_- = \{i \mid y_i^* < 0\}$ and $A_0 = \{i \mid y_i^* \leq 0\}$. Then $A_-$ ($A_0$) is the unique smallest (largest) minimizer of $F$.*

Moreover, if instead of hard-thresholding we send the min-norm point through a sigmoid, the result has the following variational inference interpretation, which lets us optimize the pipeline in eq. (3).

**Lemma 2** ([8, Thm. 3]). *Define the infinite Rényi divergence between any distributions $P$ and $Q$ over $2^V$ as $D_\infty(P \parallel Q) = \sup_{A \subseteq V} \log \left[ P(A)/Q(A) \right]$. For $P(A) \propto \exp(-F(A))$, the distribution $Q^*$ minimizing $D_\infty$ over all fully factorized distributions $Q$ is given as*

$$Q(A) = \prod_{i \in A} \sigma(-y_i^*) \prod_{i \notin A} \sigma(y_i^*),$$

*where $\sigma(u) = 1/(1 + \exp(-u))$ is the sigmoid function.*

# 3 Argmin-Differentiation of Isotonic Regression

We will first analyze a simpler problem, i.e., that of isotonic regression, defined as

$$\mathbf{y}(\mathbf{x}) = \arg\min_{\mathbf{y} \in \mathcal{O}} \frac{1}{2}\|\mathbf{y} - \mathbf{x}\|^2, \tag{8}$$

where $\mathcal{O} = \{\mathbf{y} \in \mathbb{R}^n \mid y_i \leq y_{i+1} \text{ for } i = 1, 2, \ldots, n-1\}$. The connection to our problem will be made clear in Section 4, and it essentially follows from the fact that the Lovász extension is linear on $\mathcal{O}$. In this section, we will be interested in computing the Jacobian $\partial\mathbf{y}/\partial\mathbf{x}$, i.e., in understanding how the solution $\mathbf{y}$ changes with respect to the input $\mathbf{x}$. The function is well-defined because of the strict convexity of the objective and the non-empty convex feasible set. Moreover, it can be easily computed in $O(n)$ time using the pool adjacent violators algorithm (PAVA) [26]. This is a well-studied problem in statistics, see e.g. [27]. To understand the behaviour of $\mathbf{y}(\mathbf{x})$, we will start by stating the optimality conditions of problem (8). To simplify the notation, for any $A \subseteq V$ we will define $\text{Mean}_{\mathbf{x}}(A) = \frac{1}{|A|}\sum_{i \in A} x_i$. The optimality conditions can be stated via ordered partitions $\Pi = (B_1, B_2, \ldots, B_m)$ of $V$, meaning that the sets $B_i$ are disjoint, $\cup_{j=1}^k B_j = V$, and $\Pi$ is ordered so that $1 + \max_{i \in B_j} i = \min_{i \in B_{j+1}} i$. Specifically, for any such partition we define $\mathbf{y}_{\Pi} = (\mathbf{y}_1, \mathbf{y}_2, \ldots, \mathbf{y}_m)$, where $\mathbf{y}_j = \text{Mean}_{\mathbf{x}}(B_j)\mathbf{1}_{|B_j|}$ and $\mathbf{1}_k = \{1\}^k$ is the vector of all ones. In other words, $\mathbf{y}_{\Pi}$ is defined by taking block-wise averages of $\mathbf{x}$ with respect to $\Pi$. By analyzing the KKT conditions of problem (8), we obtain the following well-known condition.

**Lemma 3** ([26]). *An ordered partition $\Pi = (B_1, B_2, \ldots, B_m)$ is optimal iff the following hold*

1. *(Primal feasibility) For any two blocks $B_j$ and $B_{j+1}$ we have*

$$Mean_{\mathbf{x}}(B_j) \leq Mean_{\mathbf{x}}(B_{j+1}). \tag{9}$$

2. *(Dual feasibility) For every block $B \in \Pi$ and each $i \in B$ define $Pre_B(i) = \{j \in B \mid j \leq i\}$. Then, the condition reads*

$$Mean_{\mathbf{x}}(Pre_B(i)) - Mean_{\mathbf{x}}(B) \geq 0. \tag{10}$$

Points where eq. (9) is satisfied with equality are of special interest, because of the following result.

**Lemma 4.** *If for some $B_j$ and $B_{j+1}$ the first condition is satisfied with equality, we can merge the two sets so that the new coarser partition $\Pi'$ will also be optimal.*

Thus, in the remaining of this section we will assume that the sets $B_j$ are chosen maximally. We will now introduce a notion that will be crucial in the subsequent analysis.

**Definition 1.** *For any block $B$, we say that $i \in B$ is a breakpoint if $Mean_{\mathbf{x}}(Pre_B(i)) = Mean_{\mathbf{x}}(B)$ and it is not the right end-point of $B$ (i.e., $i < \max_{i' \in B} i'$).*

From an optimization perspective, any breakpoint is equivalent to non-strict complementariness of the corresponding Lagrange multiplier. From a combinatorial perspective, they correspond to positions where we can refine $\Pi$ into a finer partition $\Pi'$ that gives rise to the same point, i.e., $\mathbf{y}_{\Pi} = \mathbf{y}_{\Pi'}$ (if we merge blocks using Lemma 4, the point where we merge them will become a breakpoint). We can now discuss the differentiability of $\mathbf{y}(\mathbf{x})$. Because projecting onto convex sets is a proximal operator and thus non-expansive, we have the following as an immediate consequence of Rademacher's theorem.

**Lemma 5.** *The function $\mathbf{y}(\mathbf{x})$ is 1-Lipschitz continuous and differentiable almost everywhere.*

We will denote by $\partial_{x_k}^-$ and $\partial_{x_k}^+$ the left and right partial derivatives with respect to $x_k$. For any index $k$ we will denote by $u(k)$ ($l(k)$) the breakpoint with the smallest (largest) coordinate larger (smaller) than $k$. Define it as $+\infty$ ($-\infty$) if no such point exists. Moreover, denote by $\Pi(\mathbf{z})$ the collection of indices where $\mathbf{z}$ takes on distinct values, i.e., $\Pi(\mathbf{z}) = \cup_{i=1}^n \{\{i' \in V \mid z_i = z_{i'}\}\}$.

**Theorem 1.** *Let $k$ be any coordinate and let $B \in \Pi(\mathbf{y}(\mathbf{x}))$ be the block containing coordinate $i$. Also define $B_+ = \{i \in B \mid i \geq u(k)\}$ and $B_- = \{i \in B \mid i \leq l(k)\}$. Hence, for any $i \in B$*

$$\partial_{x_k}^+(y_i) = [\![i \in B \setminus B_-]\!]/|B \setminus B_-|, \quad and \quad \partial_{x_k}^-(y_i) = [\![i \in B \setminus B_+]\!]/|B \setminus B_+|.$$

Note that all of these derivatives will agree iff there are no breakpoints, which means that the existence of breakpoints is an isolated phenomenon due to Lemma 5. In this case the Jacobian exists and has a very simple block-diagonal form. Namely, it is equal to

$$\frac{\partial \mathbf{y}}{\partial \mathbf{x}} = \Lambda(\mathbf{y}(\mathbf{x})) \equiv \texttt{blkdiag}(C_{|B_1|}, C_{|B_2|}, \ldots, C_{|B_m|}), \tag{11}$$

where $C_k = \mathbf{1}_{k \times k}/k$ is the averaging matrix with elements $1/k$. We will use $\Lambda(\mathbf{z})$ for the matrix taking block-wise averages with respect to the blocks $\Pi(\mathbf{z})$. As promised in the introduction, Jacobian multiplication $\Lambda(\mathbf{y}(\mathbf{x}))\mathbf{u}$ is linear as we only have to perform block-wise averages.

## 4 Min-Norm Differentiation

In this section we will assume that we have a function $F_{\boldsymbol{\theta}}$ parametrized by some $\boldsymbol{\theta} \in \mathbb{R}^d$ that we seek to learn. For example, we could have a mixture model

$$F_{\boldsymbol{\theta}}(A) = \sum_{j=1}^{d} \theta_j G_j(A), \tag{12}$$

for some fixed submodular functions $G_j : 2^V \to \mathbb{R}$. In this case, to ensure that the resulting function is submodular we also want to enforce $\theta_j \geq 0$ unless $G_j$ is modular. We would like to note that the discussion in this section goes beyond such models. Remember that the min-norm point is defined as

$$\mathbf{y}_{\boldsymbol{\theta}} = -\operatorname*{arg\,min}_{\mathbf{y}} f_{\boldsymbol{\theta}}(\mathbf{y}) + \frac{1}{2}\|\mathbf{y}\|^2, \tag{13}$$

where $f_{\boldsymbol{\theta}}$ is the Lovász extension of $F_{\boldsymbol{\theta}}$. Hence, we want to compute $\partial \mathbf{y}/\partial \boldsymbol{\theta}$. To make the connection with isotonic regression, remember how we evaluate the Lovász extension at $\mathbf{y}$. First, we pick a permutation $\sigma$ that sorts $\mathbf{y}$, and then evaluate it as $f_{\boldsymbol{\theta}}(\mathbf{y}) = \mathbf{f}_{\boldsymbol{\theta}}(\sigma)^T \mathbf{y}$, where $\mathbf{f}_{\boldsymbol{\theta}}(\sigma)$ is defined in eq. (6). Hence, the Lovász extension is linear on the set of all vectors that are sorted by $\sigma$. Formally, for any permutation $\sigma$ the Lovász extension is equal to $\mathbf{f}_{\boldsymbol{\theta}}(\sigma)^T \mathbf{y}$ on the *order cone*

$$\mathcal{O}(\sigma) = \{\mathbf{y} \mid y_{\sigma(n)} \leq y_{\sigma(n-1)} \leq \ldots \leq y_{\sigma(1)}\}.$$

Given a permutation $\sigma$, if we constrain eq. (13) to $\mathcal{O}(\sigma)$ we can replace $f_{\boldsymbol{\theta}}(\mathbf{y})$ by the linear function $\mathbf{f}_{\boldsymbol{\theta}}(\sigma)^T$, so that the problem reduces to

$$\mathbf{y}_{\boldsymbol{\theta}}(\sigma) = -\operatorname*{arg\,min}_{\mathbf{y} \in \mathcal{O}(\sigma)} \frac{1}{2}\|\mathbf{y} + \mathbf{f}_{\boldsymbol{\theta}}(\sigma)\|^2, \tag{14}$$

which is an instance of isotonic regression if we relabel the elements of $V$ using $\sigma$. Roughly, the idea is to instead differentiate eq. (14) with $\mathbf{f}_{\boldsymbol{\theta}}(\sigma)$ computed at the optimal point $\mathbf{y}_{\boldsymbol{\theta}}$. However, because we can choose an arbitrary order among the elements with equal values, there may be multiple permutations that sort $\mathbf{y}_{\boldsymbol{\theta}}$, and this extra choice we have seems very problematic. Nevertheless, let us continue with this strategy and analyze the resulting approximations to the Jacobian. We propose the following approximation to the Jacobian

$$\frac{\partial \mathbf{y}_{\boldsymbol{\theta}}}{\partial \boldsymbol{\theta}} \approx \widehat{J}_\sigma \equiv \underbrace{\Lambda(\mathbf{y}_{\boldsymbol{\theta}})}_{\approx \frac{\partial \mathbf{y}_{\boldsymbol{\theta}}(\sigma)}{\partial \mathbf{f}_{\boldsymbol{\theta}}(\sigma)}} \times \frac{\partial \mathbf{f}_{\boldsymbol{\theta}}(\sigma)}{\partial \boldsymbol{\theta}} = \Lambda(\mathbf{y}_{\boldsymbol{\theta}}) \times [\partial_{\theta_1}\mathbf{f}_{\boldsymbol{\theta}}(\sigma) \mid \partial_{\theta_2}\mathbf{f}_{\boldsymbol{\theta}}(\sigma) \mid \cdots \mid \partial_{\theta_d}\mathbf{f}_{\boldsymbol{\theta}}(\sigma)],$$

where $\Lambda(\mathbf{y}_{\boldsymbol{\theta}})$ is used as an approximation of a Jacobian which might not exist. Fortunately, due to the special structure of the linearizations, we have the following result that the gradient obtained using the above strategy *does not depend* on the specific permutation $\sigma$ that was chosen.

**Theorem 2.** *If $\partial_{\theta_k} F(A)$ exists for all $A \subseteq V$ the approximate Jacobians $\widehat{J}_\sigma$ are equal and do not depend on the choice of $\sigma$. Specifically, the $j$-th block of any element $i \in B \in \Pi(\mathbf{y}_{\boldsymbol{\theta}})$ is equal to*

$$\frac{1}{|B|}\partial_{\theta_j} F_{\boldsymbol{\theta}}(B \mid \{i' \mid [\mathbf{y}_{\boldsymbol{\theta}}]_{i'} < [\mathbf{y}_{\boldsymbol{\theta}}]_i\}). \tag{15}$$

*Proof sketch, details in supplement.* Remember that $\Lambda(\mathbf{y}_{\boldsymbol{\theta}})$ averages $\mathbf{f}_{\boldsymbol{\theta}}(\sigma)$ within each $B \in \Pi(\mathbf{y}_{\boldsymbol{\theta}})$. Moreover, as $\sigma$ sorts $\mathbf{y}_{\boldsymbol{\theta}}$, the elements in $B$ must be placed consecutively. The coordinates of $\mathbf{f}_{\boldsymbol{\theta}}(\sigma)$ are marginal gains (6) and they will telescope inside the mean, which yields the claimed quantity. □

**Graph cuts.** As a special, but important case, let us analyze how the approximate Jacobian looks like for a cut function (eq. (1)), in which case eq. (13) reduces to eq. (2). Let $\Pi(\mathbf{y}(\mathbf{w}, \mathbf{v})) = (B_1, B_2, \ldots, B_m)$. For any element $i \in V$ we will denote by $\eta(i) \in \{1, 2, \ldots, m\}$ the index of the block where it belongs to. Then, the approximate Jacobian $\widehat{J}$ at $\boldsymbol{\theta} = (\mathbf{w}, \mathbf{v})$ has entries

$$\widehat{\partial}_{v_j}(y_i) = [\![\eta(i) = \eta(j)]\!]/|B_{\eta(i)}|, \text{ and}$$

$$\widehat{\partial}_{w_{i,j}}(y_k) = \begin{cases} \text{sign}(y_i - y_j)\frac{1}{|B_{\eta(k)}|} & \text{if } \eta(k) = \eta(i), \text{ or} \\ \text{sign}(y_j - y_i)\frac{1}{|B_{\eta(k)}|} & \text{if } \eta(k) = \eta(j), \text{ and} \\ 0 & \text{otherwise,} \end{cases}$$

where the sign function is defined to be zero if the argument is zero. Intuitively, increasing the modular term $v_i$ by $\delta$ will increase all the coordinates $B$ in $\mathbf{y}$ that are in the same segment by $\delta/|B|$. On the other hand, increasing the weight of an edge $w_{i,j}$ will have no effect if $i$ and $j$ are already on $\mathbf{y}$ in the same segment, and otherwise it will pull the segments containing $i$ and $j$ together by increasing the smaller one and decreasing the larger one. In the supplementary we provide a `pytorch` module that executes the back propagation pass in $O(|V| + |E|)$ time in about 10 lines of code, and we also derive the approximate Jacobians for concave-of-cardinality and facility location functions.

## 5 Analysis

We will now theoretically analyze the conditions under which our approximation is correct, and then give a characterization of the *exact* directional derivative together with a polynomial algorithm that computes it. The first notion that will have implications for our analysis is that of (in)separability.

**Definition 2.** *The function $F: 2^V \to \mathbb{R}$ is said to be separable if there exists some $B \subseteq V$ such that $B \notin \{\emptyset, V\}$ and $F(V) = F(B) + F(V \setminus B)$.*

The term separable is indeed appropriate as it implies that $F(A) = F(A \cap B) + F((V \setminus B) \cap A)$ for all $A \subseteq V$ [2, Prop. 4.3], i.e., the function splits as a sum of two functions on disjoint domains. Hence, we can split the problem into two (on $B$ and $V \setminus B$) and analyze them independently. We would like to point out that separability is checkable in cubic time using the algorithm of Queyranne [28]. To simplify the notation, we will assume that we want to compute the derivative at point $\boldsymbol{\theta}' \in \mathbb{R}^d$ which results in the min-norm point $\mathbf{y}' = \mathbf{y}_{\boldsymbol{\theta}} \in \mathbb{R}^n$. We will furthermore assume that $\mathbf{y}'$ takes on unique values $\gamma_1 < \gamma_2 < \cdots < \gamma_k$ on sets $B_1, B_2, \ldots, B_k$ respectively, and we will define the chain $\emptyset = A_0 \subseteq A_1 \subseteq A_2 \subseteq \cdots \subseteq A_k = V$ by $A_j = \cup_{j'=1}^{j} B_{j'}$. A central role in the analysis will be played by the set of constraints in $\mathcal{B}(F_{\boldsymbol{\theta}})$ (see (5)) that are active at $\mathbf{y}_{\boldsymbol{\theta}}$, which makes sense given that we expect small perturbations in $\boldsymbol{\theta}'$ to result in small changes in $\mathbf{y}_{\boldsymbol{\theta}'}$ as well.

**Definition 3.** *For any submodular function $F: 2^V \to \mathbb{R}$ and any point $\mathbf{z} \in \mathcal{B}(F)$ we shall denote by $\mathcal{D}_F(\mathbf{z})$ the lattice of tight sets of $\mathbf{z}$ on $\mathcal{B}(F)$, i.e.*

$$\mathcal{D}_F(\mathbf{z}) = \{A \subseteq V \mid z(A) = F(A)\}.$$

The fact that the above set is indeed a lattice is well-known [1]. Moreover, note that $\mathcal{D}_F(\mathbf{z}) \supseteq \{\emptyset, V\}$. We will also define $\mathcal{D}' = \mathcal{D}_{F_{\boldsymbol{\theta}'}}(\mathbf{y}')$, i.e., the lattice of tight sets at the min-norm point.

### 5.1 When will the approximate approach work?

We will analyze sufficient conditions so that irrespective of the choice of $\sigma$, the isotonic regression problem eq. (14) has no breakpoints, and the left and right derivatives agree. To this end, for any $j \in \{1, 2, \ldots, k\}$ we define the submodular function $F_j: 2^{B_j}$ as $F_j(H) = F_{\boldsymbol{\theta}'}(H \mid A_{j-1})$, where we have dropped the dependence on $\boldsymbol{\theta}'$ as it will remain fixed throughout this section.

**Theorem 3.** *The approximate problem* (14) *is argmin-continuously differentiable irrespective of the chosen permutation $\sigma$ sorting $\mathbf{y}_{\boldsymbol{\theta}}$ if and only if any of the following equivalent conditions hold.*

*(a)* $\arg\min_{H \in B_j} \left[ F_j(H) - F_j(B_j)|H|/|B_j| \right] = \{\emptyset, B_j\}$.

*(b)* $\mathbf{y}'_{B_j} \in relint(\mathcal{B}(F_j))$, *i.e.* $\mathcal{D}_{F_j}(\mathbf{y}'_{B_j}) = \{\emptyset, B_j\}$, *which is only possible if $F_j$ is inseparable.*

In other words, we can equivalently say that the optimum has to lie on the interior of the face. Moreover, if $\boldsymbol{\theta} \to \mathbf{y}_{\boldsymbol{\theta}}$ is continuous[1], this result implies that the min-norm point is locally defined as averaging within the same blocks using (15), so that the approximate Jacobian is exact.

We would like to point out that one can obtain the same derivatives as the ones suggested in §4, if we perturb the KKT conditions, as done by Amos and Kolter [18]. If we use that approach, in addition to the computational challenges, there is the problem of non-uniqueness of the Lagrange multiplier, and moreover, some valid multipliers might be zero for some of the active constraints. This would render the resulting linear system rank deficient, and it is not clear how to proceed. Remember that when we analyzed the isotonic regression problem in §3 we had non-differentiability due to the exactly same reason — zero multipliers for active constraints, which in that case correspond to the breakpoints.

**Theorem 4.** *For a specific Lagrange multiplier there exists a solution to the perturbed KKT conditions derived by [18] that gives rise to the approximate Jacobians from Section 4. Moreover, this multiplier is unique if the conditions of Theorem 3 are satisfied.*

## 5.2 Exact computation

Unfortunately, computing the gradients exactly seems very complicated for arbitrary parametrizations $F_{\boldsymbol{\theta}}$, and we will focus our attention to mixture models of the form given in eq. (12). The directions $\mathbf{v}$ where we will compute the directional derivatives will have in general non-negative components $v_j$, unless $F_j$ is modular. By leveraging the theory of Shapiro [29], and exploiting the structure of both the min-norm point and the polyhedron $\mathcal{B}(F_{\mathbf{v}} \mid \mathcal{D}')$ we obtain at the following result.

**Theorem 5.** *Assume that $F_{\boldsymbol{\theta}'}$ is inseparable and let $\mathbf{v}$ be any direction so that $F_{\mathbf{v}}$ is submodular. The directional derivative $\partial \mathbf{y}/\partial \theta_j$ at $\boldsymbol{\theta}'$ in direction $\mathbf{v}$ is given by the solution of the following problem.*

$$
\begin{aligned}
&\underset{\mathbf{d}}{\text{minimize}} \quad \frac{1}{2}\|\mathbf{d}\|^2, \\
&\text{subject to } \mathbf{d} \in \mathcal{B}(F_{\mathbf{v}} \mid \mathcal{D}'), \text{ and} \\
&\qquad\qquad d(B_j) = F_{\mathbf{v}}(A_j) \text{ for } j \in \{1, 2, \dots, k\}.
\end{aligned}
\tag{16}
$$

First, note that this is again a min-norm problem, but now defined over a reduced lattice $\mathcal{D}'$ with $k$ additional equality constraints. Fortunately, due to these additional equalities, we can split the above problem into $k$ *separate min-norm problems*. Namely, for each $j \in \{1, 2, \dots, k\}$ collect the lattice of tight sets that intersect $B_j$ as $\mathcal{D}'_j = \{H \cap B_j \mid H \in \mathcal{D}'\}$, and define the function $G_j \colon 2^{B_j} \to \mathbb{R}$ as $G_j(A) = F_{\mathbf{v}}(A \mid A_{j-1})$, where note that the parameter vector $\boldsymbol{\theta}$ is taken as the direction $\mathbf{v}$ in which we want to compute the derivative. Then, the block of the optimal solution of problem (16) corresponding to $B_j$ is equal to

$$
\mathbf{d}^*_{B_j} = \underset{\mathbf{y}_j \in \mathcal{B}(G_j \mid \mathcal{D}'_j)}{\arg\min} \frac{1}{2}\|\mathbf{y}_j\|^2,
\tag{17}
$$

which is again a min-norm point problem where the base polytope is defined using the lattice $\mathcal{D}'_j$. We can then immediately draw a connection with the results from the previous subsection.

**Corollary 1.** *If all latices are trivial, the solution of (17) agrees with the approximate Jacobian (15).*

**How to solve problem (16)?** Fortunately, the divide-and-conquer algorithm of Groenevelt [30] can be used to find the min-norm point over arbitrary lattices. To do this, we have to compute for each $i \in B_j$ the unique smallest set $H_i^*$ in $\arg\min_{H_j \ni i} F_j(H_j) - y'(H_j)$, which can be done using submodular minimization after applying the reduction of Schrijver [31].

To highlight the difference with the approximation from section 4, let us consider a very simple case.

**Lemma 6.** *Assume that $G_j$ is equal to $G_j(A) = [\![i \in A]\!]$ for some $i \in B_j$. Then, the directional derivative is equal to $\mathbf{1}_{|D|}/|D|$ where $D = \{i' \mid i \in H_{i'}^*\}$.*

Hence, while the approximate directional derivative would average over all elements in $B_j$, the true one averages only over a subset $D \subseteq B_j$ and is possibly sparser. Lemma 6 gives us the exact directional derivatives for the graph cuts, as each component $G_j$ will be either a cut function on

two vertices, or a function of the form in Lemma 6. In the first case the directional derivative is zero because $\mathbf{0} \in \mathcal{B}(G_j) \subseteq \mathcal{B}(G_j \mid \mathcal{D}'_j)$. In the second case, we can can either solve exactly using Lemma 6 or use a more sophisticated approximation, generalizing the result from [32] — given that $F_j$ is separable over $2^{B_j}$ iff the graph is disconnected, we can first separate the graph into connected components, and then take averages within each connected component instead of over $B_j$.

## 5.3 Structured attention and constraints

Recently, there has been an interest in the design of structured attention mechanisms, which, as we will now show, can be derived and furthermore generalized using the results in this paper. The first mechanism is the *sparsemax* of Martins and Astudillo [33]. It takes as input a vector and projects it to the probability simplex, which is the base polytope corresponding to $G(A) = \min\{|A|, 1\}$. Concurrently with this work, Niculae and Blondel [32] have analyzed the following problem

$$\mathbf{y}^* = \min_{\mathbf{y} \in \mathcal{B}(G)} f(\mathbf{y}) + \frac{1}{2}\|\mathbf{y} - \mathbf{z}\|^2, \tag{18}$$

for the special case when $\mathcal{B}(G)$ is the simplex and $f$ is the Lovász extension of one of two specific submodular functions. We will consider the general case when $G$ can be any concave-of-cardinality function and $F$ is an arbitrary submodular function. Note that, if either $f(\mathbf{y})$ or the constraint were not present in problem (18), we could have simply leveraged the theory we have developed so far to differentiate it. Fortunately, as done by Niculae and Blondel [32], we can utilize the result of Yu [34] to significantly simplify (18). Namely, because projection onto $\mathcal{B}(G)$ preserves the order of the coordinates [35, Lemma 1], we can write the optimal solution $\mathbf{y}^*$ to (18) as

$$\mathbf{y}^* = \min_{x \in \mathcal{B}(G)} \frac{1}{2}\|\mathbf{y} - \mathbf{y}'\|, \text{ where } \mathbf{y}' = \arg\min_{\mathbf{y}} f(\mathbf{y}) + \frac{1}{2}\|\mathbf{y} - \mathbf{z}\|^2.$$

We can hence split problem (18) into two subtasks — first, compute $\mathbf{y}'$ and then project it onto $\mathcal{B}(G)$. As each operation can reduces to a minimum-norm problem, we can differentiate each of them separately, and thus solve (18) by *stacking* two submodular layers one after the other.

## 6   Experiments

We consider the image segmentation tasks from the introduction, where we are given an RGB image $\mathbf{x} \in \mathbb{R}^{n \times 3}$ and are supposed to predict those pixels $\mathbf{y} \in \{0, 1\}^n$ containing the foreground object. We used the Weizmann horse segmentation dataset [36], which we split into training, validation and test splits of sizes $180$, $50$ and $98$ respectively. The implementation was done in `pytorch`[2], and to compute the min-norm point we used the algorithm from [37]. To

|  |  | CNN |  | CNN+GC |  |
|---|---|---|---|---|---|
|  |  | Mean | Std. Dev. | Mean | Std. Dev. |
| Accuracy |  | 0.8103 | 0.1391 | **0.9121** | 0.1034 |
| NLL |  | 0.3919 | 0.1911 | **0.2681** | 0.2696 |
| # Fg. Objs. |  | 96.9 | 65.8 | **25.3** | 30.6 |

Figure 1: Test set results. We see that incorporating a graph cut solver improves both the accuracy and negative log-likelihood (NLL), while having consistent segmentations with fewer foreground objects.

make the problem more challenging, at training time we randomly selected and revealed only $0.1\%$ of the training set labels. We first trained a convolutional neural network with two hidden layers that directly predicts the per-pixel labels, which we refer to as CNN. Then, we added a second model, which we call CNN+GC, that has the same architecture as the first one, but with an additional graph cut layer, whose weights are parametrized by a convolutional neural network with one hidden layer. Details about the architectures can be found in the supplementary. We train the models by maximizing the log-likelihood of the revealed pixels, which corresponds to the variational bi-level strategy (eq. (3)) due to Lemma 2. We trained using SGD, Adagrad [38] and Adam [39], and chose the model with the best validation score. As evident from the results presented in Section 6, adding the discrete layer improves not only the accuracy (after thresholding the marginals at $0.5$) and log-likelihood, but it gives more coherent results as it makes predictions with fewer connected components (i.e., foreground objects). Moreover, if we have a look at the predictions themselves in Figure 2, we can observe that the optimization layer not only removes spurious predictions, but there is is also a qualitative difference in the marginals as they are spatially more consistent.

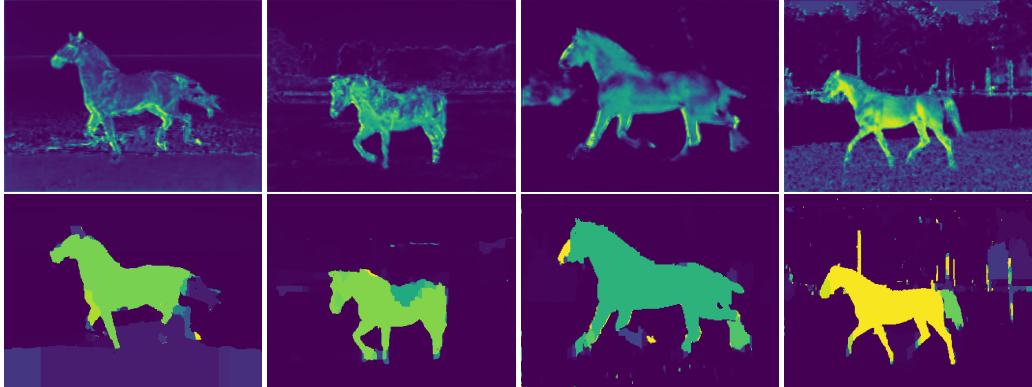

Figure 2: Comparison of results from both models on four instances from the test set (up: CNN, down: CNN+GC). We can see that adding the graph-cut layers helps not only quantitatively, but also qualitatively, as the predictions are more spatially regular and vary smoothly inside the segments.

# 7    Conclusion

We have analyzed the sensitivity of the min-norm point for parametric submodular functions and provided both a very easy-to-implement practical approximate algorithm for general objectives, and strong theoretical result characterizing the true directional derivatives for mixtures. These results allow the use of submodular minimization inside modern deep architectures, and they are also immediately applicable to bi-level variational learning of log-supermodular models of arbitrarily high order. Moreover, we believe that the theoretical results open the new problem of developing algorithms that can compute not only the min-norm point, but also solve for the associated derivatives.

**Acknowledgements.**   The research was partially supported by ERC StG 307036 and a Google European PhD Fellowship.

## Footnotes

[1] For example if the correspondence $\boldsymbol{\theta} \twoheadrightarrow \mathcal{B}(F_{\boldsymbol{\theta}})$ is hemicontinuous due to Berge's theorem.

[2]The code will be made available at `https://www.github.com/josipd/nips-17-experiments`.

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
