[Supplementary Material · appendix.pdf]

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

## A    Approximate Jacobians for certain function classes

**Cardinality functions.**    Remember that these functions are of the form $F(A) = h(|A|)$ for some concave $h$ satisfying $h(0) = 0$. Because we only evaluate $h$ at the integers $\{1, 2, \ldots, n\}$, we can parametrize it by $h(k) = \sum_{i=1}^{k} \theta_i$ for some reals $\theta_1 \geq \theta_2 \geq \cdots \geq \theta_n$ (this condition is equivalent to the concavity of $h$). Then, $\partial_{\theta_k} \mathbf{f}_\sigma$ is equal to one on coordinate $\sigma(k)$ and zero elsewhere. Hence, for any block $B \in \Pi(\mathbf{y_\theta})$, the column of the approximate Jacobian corresponding to $\partial_{\theta_k}$ will have values that are identical and equal to $[\![\sigma(k) \in B]\!]/|B|$. Again, backpropagation takes linear time.

**Facility location.**    Consider functions of the form $F(A) = \max_{i \in A} w_i$ for some non-negative weights $w_i$. For any weight $w_k$ the approximate derivative $\widehat{\partial}_{w_k}$ is non-zero only for the block $B$ is containing $\sigma(k)$ and the block $B_-$ succeeding it. It can be again computed in linear time.

## B    Proofs

**Definition 4.** *Denote by* $\mathrm{Sym}(V)$ *the set of all permutations* $\sigma \colon V \to V$ *of* $V$.

*Proof of lemma 4.* The first condition (eq. (9)) is obviously satisfied after merging, so that we only have to prove that dual feasibility (eq. (10)) will not be violated. Denote the new merged block by $B = B_j \cup B_{j+1}$. First, note that by assumption $\mathrm{Mean}_\mathbf{x}(B) = \mathrm{Mean}_\mathbf{x}(B_j) = \mathrm{Mean}_\mathbf{x}(B_{j+1})$. For any $i \in B_j$ we will have $\mathrm{Mean}_\mathbf{x}(\mathrm{Pre}_{B_j}(i)) = \mathrm{Mean}_\mathbf{x}(\mathrm{Pre}_B(i))$, so the dual feasibility condition is satisfied from the assumption that the original partition was optimal. On the other hand, for $i \in B_{j+1}$

$$\mathrm{Mean}_\mathbf{x}(B_j \cup \mathrm{Pre}_{B_{j+1}}(i)) - \mathrm{Mean}_\mathbf{x}(B_j \cup B_{j+1})$$
$$= \alpha \mathrm{Mean}_\mathbf{x}(B_j) + (1 - \alpha)\mathrm{Mean}_\mathbf{x}(\mathrm{Pre}_{B_{j+1}}(i)) - \mathrm{Mean}_\mathbf{x}(B_j \cup B_{j+1}) \text{ for } \alpha = \frac{|B_j|}{|B_j| + |B_{j+1}|}$$
$$= \alpha \big[ \mathrm{Mean}_\mathbf{x}(B_j) - \mathrm{Mean}_\mathbf{x}(B_j) \big] + (1 - \alpha)\big[ \mathrm{Mean}_\mathbf{x}(\mathrm{Pre}_{B_{j+1}}(i)) - \mathrm{Mean}_\mathbf{x}(B_{i+1}) \big]$$
$$\geq 0.$$

$\square$

*Proof of theorem 1.* We will show how to compute the derivative in direction $\partial_{x_k}^+$, as the other case follows analogously. Let us split $B$ into $B_-$ and $B'_+ = B \setminus B_- = \{i_1 + 1, i_1 + 2, \ldots, i_1 + m\}$, and consider the partition $\Pi'$ formed by replacing $B$ with $\{B_-, B'_+\}$. We will show that the above partition is optimal when we increase $x_k$ by any sufficiently small perturbation $\varepsilon > 0$. Then, the claim immediately follows, because $\mathbf{y}(\mathbf{x})$ is defined by taking averages inside an optimal partition. We thus have to show that both of the conditions in lemma 3 will hold for $\Pi'$. The first condition obviously holds as the elements of $\Pi(\mathbf{y}(\mathbf{x}))$ correspond to different values, so there is some non-zero gap between the averages of consecutive blocks. Let us denote by $\mathbf{x}_\varepsilon$ the perturbed point, i.e. $x_{\varepsilon,j} = x_j + [\![j = k]\!]\varepsilon$. Because $B_-$ ends with a breakpoint and $k \notin B_-$, we have that

$$\mathrm{Mean}_{\mathbf{x}_\varepsilon}(B_-) = \mathrm{Mean}_\mathbf{x}(B) = \mathrm{Mean}_\mathbf{x}(B'_+), \tag{19}$$

which means that the second condition also holds for $B_-$ as none of its points are perturbed, so that the averages stay the same. On the other hand, for any $i \in B'_+$

$$\mathrm{Mean}_\mathbf{x}(\mathrm{Pre}_B(i)) = (1 - \alpha)\mathrm{Mean}_\mathbf{x}(B_-) + \alpha \mathrm{Mean}_\mathbf{x}(\mathrm{Pre}_{B'_+}(i)) \text{ for } \alpha = \frac{i - i_1}{|B|} \in (0, 1].$$

Moreover, if we subtract $\mathrm{Mean}_\mathbf{x}(B)$ from both sides of the above equality we have (from eq. (19))

$$\mathrm{Mean}_\mathbf{x}(\mathrm{Pre}_B(i)) - \mathrm{Mean}_\mathbf{x}(B) = \alpha \big[ \mathrm{Mean}_\mathbf{x}(\mathrm{Pre}_{B'_+}(i)) - \mathrm{Mean}_\mathbf{x}(B'_+) \big]. \tag{20}$$

Hence, for any $i \in B'_+$

$$\mathrm{Mean}_{\mathbf{x}_\varepsilon}(\mathrm{Pre}_{B'_+}(i)) - \mathrm{Mean}_{\mathbf{x}_\varepsilon}(B'_+) = \mathrm{Mean}_{\mathbf{x}}(\mathrm{Pre}_{B'_+}(i)) + [\![i \geq k]\!]\frac{\varepsilon}{i - i_1} - \mathrm{Mean}_{\mathbf{x}}(B'_+) - \frac{\varepsilon}{|B'_+|}$$

$$= \mathrm{Mean}_{\mathbf{x}}(\mathrm{Pre}_{B'_+}(i)) - \mathrm{Mean}_{\mathbf{x}}(B'_+) + \varepsilon\Big[\frac{[\![i \geq k]\!]}{i - i_1} - \frac{1}{m}\Big]$$

$$= \frac{1}{\alpha}\underbrace{\big[\mathrm{Mean}_{\mathbf{x}}(\mathrm{Pre}_B(i)) - \mathrm{Mean}_{\mathbf{x}}(B)\big]}_{D \geq 0} + \varepsilon\underbrace{\Big[\frac{[\![i \geq k]\!]}{i - i_1} - \frac{1}{m}\Big]}_{F},$$

where in the last equality we used eq. (20). What remains is to show that the above quantity is always positive except at the end-point. First, if $i \geq k$, then the $F$ is strictly positive for all points except for the end-point $i = i_1 + m$ in which case it evaluates to zero. Now, let $i < k$. Then, because $i > l(k)$ it can-not be a breakpoint, so that $D > 0$. Then, for all $\varepsilon$ smaller than some sufficiently small $\delta$ the quantity $F$ will also be positive, which implies dual feasibility for $B'_+$ and completes the proof. $\square$

**Definition 5.** *We define $\Sigma(\mathbf{z})$ as the set of permutations that sort $\mathbf{z}$ in descending order, or formally stated*

$$\Sigma(\mathbf{z}) = \{\sigma \in Sym(V) \mid z_{\sigma(1)} \geq z_{\sigma(2)} \geq \ldots \geq z_{\sigma(n)}\}.$$

*Proof of theorem 2.* We will show that the $j$-th column of the approximate Jacobian, equal to $\Lambda(\mathbf{y}_{\boldsymbol{\theta}})\partial_{\theta_j}\mathbf{f}_{\boldsymbol{\theta}}(\sigma)$ does not depend on the choice of $\sigma$. First, note that the $\sigma(i)$-th coordinate of $\partial_{\theta_j}\mathbf{f}_{\boldsymbol{\theta}}(\sigma)$ is equal to

$$[\partial_{\theta_j}\mathbf{f}_{\boldsymbol{\theta}}(\sigma)]_{\sigma(i)} = \partial_{\theta_j}F_{\boldsymbol{\theta}}(\{\sigma(i)\} \mid \{\sigma(1), \sigma(2), \ldots, \sigma(i-1)\})$$
$$= \partial_{\theta_j}F_{\boldsymbol{\theta}}(\{\sigma(1), \sigma(2), \ldots, \sigma(i)\}) - \partial_{\theta_j}F_{\boldsymbol{\theta}}(\{\sigma(1), \sigma(2), \ldots, \sigma(i-1)\}). \tag{21}$$

Let $B$ be the block where $i$ belongs to, i.e. $B = \{i' \mid [\mathbf{y}_{\boldsymbol{\theta}}]_{i'} = [\mathbf{y}_{\boldsymbol{\theta}}]_i\}$. Because we chose a partition $\sigma$ that sorts $\mathbf{y}_{\boldsymbol{\theta}}$, all elements in $B$ must have their elements placed consecutively under $\sigma$, i.e. it there must exist some $i_1$ so that $B = \{\sigma(i_1 + 1), \sigma(i_2 + 2), \ldots, \sigma(i_1 + |B|)\}$. Then

$$\mathrm{Mean}_{\partial_{\theta_j}\mathbf{f}_{\boldsymbol{\theta}}(\sigma)}(B)$$

$$= \frac{1}{|B|}\sum_{i=i_1+1}^{i_1+|B|}\partial_{\theta_j}F_{\boldsymbol{\theta}}(\{\sigma(i)\} \mid \{\sigma(1), \ldots, \sigma(i_1)\})$$

$$= \frac{1}{|B|}\Big[\sum_{i=i_1+1}^{i_1+|B|}\partial_{\theta_j}F_{\boldsymbol{\theta}}(\{\sigma(1), \sigma(2), \ldots, \sigma(i)\}) - \partial_{\theta_j}F_{\boldsymbol{\theta}}(\{\sigma(1), \sigma(2), \ldots, \sigma(i-1)\})\Big]$$

$$= \frac{\partial_{\theta_j}}{|B|}F_{\boldsymbol{\theta}}(\{\sigma(1), \ldots, \sigma(i_1 + |B|)\} \mid \{\sigma(1), \ldots, \sigma(i_1)\})$$

$$= \frac{\partial_{\theta_j}}{|B|}F_{\boldsymbol{\theta}}(B \mid \{i' \mid [\mathbf{y}_{\boldsymbol{\theta}}]_{i'} < [\mathbf{y}_{\boldsymbol{\theta}}]_i\}),$$

where the third equality follows from the telescoping of the summands. $\square$

*Proof of theorem 3.* From theorem 1 we can see that the left and right derivatives will agree if and only if each block $B_j$ has no of the chosen order cone. We will refer to such blocks as being *unbreakable*. The first condition then follows from the following lemma.

**Lemma 7.** *Block $B_j$ is unbreakable if and only if there is no $H \notin \{\emptyset, B_j\}$ satisfying*

$$F_j(H)/|H| = F_j(B_j)/|B_j|.$$

*Moreover, this condition is equivalent to $\arg\min_{H \in B_j}\big[F_j(H) - F_j(V)|H|/|B_j|\big] = \{\emptyset, B_j\}$.*

*Proof.* Let w.l.o.g. $B_j = \{1, 2, \ldots, k\}$ and let $\sigma \colon \{1, 2, \ldots, k\} \to \{1, 2, \ldots, k\}$ be the ordering the chosen permutation makes on $B_j$. Then, at the $i$-th position inside $B_j$, the linearization has

$$[\mathbf{f}(\sigma)]_i = F_j(\{\sigma(i)\} \mid \{\sigma(1), \sigma(2), \ldots, \sigma(i-1)\}).$$

Then, due to the telescoping sums we have

$$\text{Mean}_{\mathbf{f}(\sigma)}(\text{Pre}_i(B_j)) = F(\{\sigma(1), \sigma(2), \ldots, \sigma(i)\})/i,$$

with the special case

$$\text{Mean}_{\mathbf{f}(\sigma)}(\text{Pre}_k(B_j)) = F(B_j)/k = F(B)/|B_j|.$$

Now, by definition, $B_j$ is unbreakable if and only if the above two quantities are never equal for $i < k$ for any possible permutation $\sigma$, which is exactly what we had to show. □

The second condition will follow as a corollary of the following well-known result characterizing the min-norm point.

**Lemma 8.** *For each $B_j$ we have that $\mathbf{y}'_{B_j} = F_j(B_j)/|B_j|$.*

*Proof.* From [1][Theorem 9.1] we know that $y'(A_j) = F_{\boldsymbol{\theta}'}(A_j)$ for each $j \in \{1, \ldots, k\}$. Subtracting the equalities for $j$ and $j - 1$ we have

$$y(B_j) = F(A_j) - F(A_{j-1}) = F_j(B_j).$$

Now, the result follows because we have assumed that all elements in $B_j$ are equal. □

**Corollary 2.** *Block $B_j$ is unbreakable iff $\mathcal{D}_{F_j}(\mathbf{y}'_{B_j}) = \{\emptyset, B_j\}$.*

*Proof.* Assume there exists some other set $H \subseteq B_j$ that is tight. This implies that $\mathbf{y}'_{B_j}(H) = F_j(H)$. Then, lemma 8 implies that

$$F_j(B_j)|H|/|B_j| = F_j(H),$$

which is exactly equal to the first condition. □

The fact that the the base polytope has any point in the relative interior being equivalent to inseparability is well-known, see e.g. [2, Prop. 4.6] or [1, Thm. 3.36]. □

*Proof of theorem 4.* Remember that the min-norm point is defined as

$$\arg\min_{\mathbf{y} \in \mathbb{R}^n} \frac{1}{2}\|\mathbf{y}\|^2$$

subject to $y(A) \leq \sum_{j=1}^d \theta_j F_j(A)$ for all $A \subsetneq V$ and $y(V) = \sum_{j=1}^d \theta_j F_j(V)$. Introducing multipliers $\lambda_A$ for the inequality constraints and $\mu$ for the equalities we arrive at the Lagrangian

$$\mathcal{L}(\mathbf{y}, \boldsymbol{\lambda}, \mu) = \frac{1}{2}\|\mathbf{y}\|^2 + \sum_{A \subsetneq V} \lambda_A(y(A) - F_{\boldsymbol{\theta}}(A)) + \mu(y(V) - F(V)).$$

Then, setting $\partial_{y_i}\mathcal{L}$ to zero yields the condition

$$\mu + \sum_{A \ni i} \lambda_A = -y_i. \tag{22}$$

Moreover, due to the complementary slackness condition can have non-zero multipliers only for the tight constraints. As $\mathbf{y}'$ is the min-norm point for $F_{\boldsymbol{\theta}'}$, we know that the sets $A_1, A_2, \ldots, A_k = V$ are all tight (see lemma 8). Remember that $\mathbf{y}'$ took values $\gamma_1 < \gamma_2 < \cdots < \gamma_k$ on sets $B_1, B_2, \ldots, B_k$ respectively. We suggest the following family of multipliers

$$\mu = \gamma_k = -\max_i y_i$$

$$\lambda_A = \begin{cases} \gamma_{j+1} - \gamma_j \geq 0 & \text{if for } A = A_j \text{ for } j \in \{1, 2, \ldots, k-1\}, \text{ or} \\ 0 & \text{otherwise.} \end{cases}$$

They indeed satisfy the condition in eq. (22) because for every $i \in B_j$ we have that

$$\mu + \sum_{A \ni i} \lambda_A = \mu + \sum_{j'=j}^{k-1} \lambda_{A_{j'}} = \gamma_k - \sum_{j'=j}^{k-1} \gamma_{j'+1} - \gamma_{j'} = -\gamma_{j'} = -y_i.$$

The fact that these multipliers are unique will easily follow if we show that the conditions imply that $A_1, A_2, \ldots, A_k$ are the only tight sets. Then, it can be easily seen that the only multipliers satisfying eq. (22) are those we have just suggested, by doing an induction on $j$ starting from $j = k$ (corresponding to $\nu$) and going backwards. For the sake of contradiction, assume that the conditions hold and that there exists some $H$ different from the members of the chain that is tight, i.e. $y'(H) = F_{\theta'}(H)$ and $A_{j-1} \subsetneq H \subsetneq A_j$. Then, if we define $H' = H \setminus A_{j-1} \notin \{\emptyset, B_j\}$, we have that

$$
\begin{aligned}
F_j(H') &= F_{\theta'}(H' \cup A_{j-1}) - F_{\theta'}(A_{j-1}) \\
&= F_{\theta'}(H) - F_{\theta'}(A_j) \\
&= y'(H) - F_{\theta'}(A_j) \\
&= y'(H') - |H'| F_{\theta'}(B_j)/|B_j|,
\end{aligned}
$$

where the last step follows from lemma 8. However, this means that $H'$ violates the assumption condition for $F_j$, which is a contradiction to our assumption.

Create the matrix $G = \{0, 1\}^{(2^n - 1) \times n}$ that has the indicator vectors of the inequality constraints as its rows. Similarly, create the vector $\mathbf{h} \in \mathbb{R}^{2^n - 1}$ with elements $F(A)$, and define $b = F(V)$. Let us first compute $\partial \mathbf{y}/\partial b$, for which we have to solve the following system

$$
\begin{aligned}
d\mathbf{y} + G^T d\boldsymbol{\lambda} + \mathbf{1} d\mu &= \mathbf{0} \\
\mathtt{diag}(\boldsymbol{\lambda}^*) G d\mathbf{y} + \mathtt{diag}(G\mathbf{y} - \mathbf{h}) d\boldsymbol{\lambda} &= \mathbf{0} \\
\mathbf{1}^T d\mathbf{y} &= 1
\end{aligned}
$$

We suggest the solution $d\mathbf{y} = \mathbf{1}_{B_k}/|B_k|, d\mu = 1/|B_k|$ and $d\boldsymbol{\lambda}$ to be zero except coordinate $d\boldsymbol{\lambda}_{A_{k-1}} = -1/|B_k|$, which can be easily seen to satisfy the conditions. Namely, the second condition translates to $\mathbf{1}_{A_j}^T d\mathbf{y} = 0$ for $j = 1, 2, \ldots, k-1$, which is true at $\mathbf{y}$ is non-zero only on $B_k$. The third condition is obvious, while the first one reads

$$
d\mathbf{y}_i = -\sum_{A \ni i} d\boldsymbol{\lambda}_A + d\mu = -1/|B_k| [\![i \in A_{k-1}]\!] + 1/|B_k| = 1/|B_k| [\![i \notin A_{k-1}]\!] = 1/|B_k| [\![i \in B_k]\!].
$$

Now, let us compute $\partial \mathbf{y}/\partial \mathbf{h}$, for which we have to solve the following system

$$
\begin{aligned}
d\mathbf{y} + G^T d\boldsymbol{\lambda} + \mathbf{1} d\mu &= \mathbf{0} \\
\mathtt{diag}(\boldsymbol{\lambda}) G d\mathbf{y} + \mathtt{diag}(G\mathbf{y} - \mathbf{h}) d\boldsymbol{\lambda} &= \mathtt{diag}(\boldsymbol{\lambda}) \\
\mathbf{1}^T d\mathbf{y} &= 0
\end{aligned}
$$

As Ansatz we suggest $d\mu = 0$ and only the following to non-zero

$$
d\mathbf{y}_{i,A} = \begin{cases} +1/|B_j| & \text{if } i \in B_j \text{ and } A = A_j \text{ for } j < k, \text{ or} \\ -1/|B_{j+1}| & \text{if } i \in B_{j+1} \text{ and } A = A_j \text{ for } j < k, \text{ or} \\ 0 & \text{otherwise,} \end{cases}
$$

and

$$
d\boldsymbol{\lambda}_{A',A} = \begin{cases} -1/|B_j| & \text{if } A' = A = A_j \text{ and } j < k, \text{ or} \\ +1/|B_{j+1}| + 1/|B_j| & \text{if } A' = A_j \text{ and } A = A_{j+1}, \text{ if } j < k, \text{ or} \\ 0 & \text{otherwise.} \end{cases}
$$

Note that the third condition is immediately satisfied for $d\mathbf{y}$, as every column sums to zero. The third condition for those rows of $d\boldsymbol{\lambda}$ corresponding to $A_1, A_2, \ldots, A_{k-1}$ reads

$$
\sum_{i \in A_j} d\mathbf{y}_{A', A_j} = [\![A = A_j]\!],
$$

which is satisfied because the $A_j$-th column of $d\mathbf{y}$ is $+1/|B_j|$ on $B_j$ and we do not sum over its other elements which are in $B_{j+1}$. Finally, the first condition is non-vacuous only for the non-zero columns $A_1, A_2, \ldots, A_{k-1}$, in which case it also holds because we have

$$
\begin{aligned}
d\mathbf{y}_{i,A_j} &= -\sum_{A \ni i} d\boldsymbol{\lambda}_{A,A_j} \\
&= [\![i \in A_j]\!]/|B_j| - (1/|B_j| + 1/|B_{j+1}|)[\![i \in A_{j+1}]\!] \\
&= [\![i \in B_j]\!]/|B_j| - [\![i \in B_{j+1}]\!]/|B_{j+1}|.
\end{aligned}
$$

Then, if these were the actual true Jacobians, for any parameter $\theta_j$ and output $i \in B_j$ we have

$$
\begin{aligned}
\frac{\partial y_i}{\partial \theta_j} &= \sum_{A \subseteq V} \frac{\partial y_i}{\partial F_\theta(A)} \times \frac{\partial F_\theta(A)}{\theta_j} \\
&= \sum_{j=1}^{k-1} \frac{\partial y_i}{\partial F_\theta(A_j)} \times \frac{\partial F_\theta(A_j)}{\theta_j} \\
&= \underbrace{\frac{\partial y_i}{\partial F_\theta(A_{j-1})}}_{-1/|B_j|} \times \frac{\partial F_\theta(A_j)}{\theta_j} + \underbrace{\frac{\partial y_i}{\partial F_\theta(A_j)}}_{1/|B_j|} \times \frac{\partial F_\theta(A_j)}{\theta_j} = \frac{\partial \theta_j}{|B_j|} F_\theta(B_j).
\end{aligned}
$$

$\square$

*Proof of corollary 1.* If the lattices are trivial problem (17) reduces to

$$
\min_{\mathbf{y}_k \, : \, \mathbf{y}_k^T \mathbf{1} = G_j(B_j)} \|\mathbf{y}_j\|^2,
$$

which has as a solution the constant vector $\mathbf{y}_k^*$ with coordinates all equal to

$$
G_j(B_j)/|B_j| = F_\mathbf{v}(B_j \mid A_{j-1})/|B_j|.
$$

$\square$

*Proof of theorem 5.* First, because the function $\boldsymbol{\theta} \to \mathbf{y}_\boldsymbol{\theta}$ is continuous under the assumptions ([29, Lem. 2.1]), we know that in a neighbourhood of $\boldsymbol{\theta}'$ no inactive constraints can activate. Hence, we can focus on the active constraints only.

**Lemma 9.** *The four assumptions made in [29] hold whenever $F_{\boldsymbol{\theta}'}$ is inseparable.*

*Proof.* We show below why each one holds.

1. We need a neighbourhood $N_{\boldsymbol{\theta}'}$ of $\boldsymbol{\theta}'$ so that for all $\boldsymbol{\theta}'' \in N_{\boldsymbol{\theta}'}$ the base polyhedra are bounded. Obviously holds as $\mathcal{B}(F_\boldsymbol{\theta}) = \sum_{j=1}^m w_j \mathcal{B}(F_j)$.

2. We have already discussed the uniqueness of the min-norm point.

3. The Mangasarian-Fromovitz condition is implied by Slater's condition, which in case holds whenever the function is inseparable [1][Thm. 3.36].

4. Clearly follows from the strong convexity of the objective.

$\square$

Let us define the shorthands $\mathcal{D}' = \mathcal{D}_{F_{\boldsymbol{\theta}'}}(\mathbf{y}')$ and $\Sigma(\mathbf{v}) = \mathcal{B}(F_\mathbf{v} \mid \mathcal{D}')$. It is easy to see that our definition of $\Sigma(\mathbf{v})$ agrees with that in [29][(2.11)] because the constraints are linear in both $\mathbf{y}$ and $\boldsymbol{\theta}$, and there is no term containing both $\mathbf{y}$ and $\boldsymbol{\theta}$. Namely, each constraint is of the form

$$
g_A(\mathbf{y}, \boldsymbol{\theta}) = \mathbf{y}^T \mathbf{1}_A - \sum_{j=1}^m \theta_j F_j(A),
$$

we have its linear approximation

$$
\alpha_A(\mathbf{u}, \mathbf{v}) = \mathbf{u}^T \nabla_\mathbf{y} g_A(\mathbf{y}', \boldsymbol{\theta}') + \mathbf{v}^T \nabla_\boldsymbol{\theta} g_A(\mathbf{y}', \boldsymbol{\theta}') = \mathbf{u}^T \mathbf{1}_A - \sum_{j=1}^m v_j F_j(A) = u(A) - F_\mathbf{v}(A).
$$

Then, the equivalence is clear given the definition [29][(2.11)]

$$
\Sigma(\mathbf{v}) = \{\mathbf{u} \in \mathbb{R}^n \mid \alpha_A(\mathbf{u}, \mathbf{v}) \le 0 \text{ for all } A \in \mathcal{D}'\} \cap \{\mathbf{u} \in \mathbb{R}^n \mid \alpha_V(\mathbf{u}, \mathbf{v}) = 0\}.
$$

**Lemma 10.** *A point* $\mathbf{u} \in \Sigma(\mathbf{v})$ *belongs to* $\overline{\Sigma}(\mathbf{v}) = \arg\min_{\mathbf{u} \in \Sigma(\mathbf{v})} \mathbf{u}^T \mathbf{y}'$ *iff it satisfies*

$$u(A_j) = F_{\mathbf{v}}(A_j) \ \ for \ j \in \{1, 2, \ldots, k\}.$$

*Proof.* First, note that $\mathbf{y}_{\boldsymbol{\theta}}^T : V \to \mathbb{R}$ is monotone non-decreasing on $\mathcal{D}'$ because for any $i \in B_i$ we have that $\mathrm{dep}(\mathbf{y}', i) \subseteq A_i$ [1][Theorem 9.1 (iii)], and the lattice we consider is exactly the one generated from the tight sets of the min-norm point. Then, the result follows from [1][Theorem 3.15]. $\qquad\square$

Now, because the Lagrangian

$$\mathcal{L}(\mathbf{y}, \boldsymbol{\theta}, \boldsymbol{\lambda}, \nu) = \frac{1}{2}\|\mathbf{y}\|^2 + \sum_{A \in \mathcal{D}'} \lambda_A(y(A) - \sum_{j=1}^{d} \theta_j F_j(A)) + \nu(y(V) + \sum_{j=1}^{d} \theta_j F_j(V))$$

has no terms containing both $\mathbf{y}$ and $\boldsymbol{\theta}$ we have that (defined as (3.3) in [29])

$$\xi_{\boldsymbol{\lambda}}(\mathbf{u}, \mathbf{v}) = \frac{1}{2}\mathbf{u}^T \underbrace{\nabla_{\mathbf{yy}}^2 \mathcal{L}}_{I} \mathbf{u} + \mathbf{u}^T \underbrace{\nabla_{\mathbf{y}\boldsymbol{\theta}}^2 \mathcal{L}}_{0} \mathbf{v} + \frac{1}{2}\mathbf{v}^T \underbrace{\nabla_{\mathbf{yy}}^2 \mathcal{L}}_{0} \mathbf{v} = \frac{1}{2}\|\mathbf{u}\|^2,$$

so that the function is *independent* of its second argument. Then, $\zeta_{\mathbf{v},\mathbf{h}}(\mathbf{u})$ (defined as (4.10) in [29]) reduces to

$$\zeta_{\mathbf{v},\mathbf{h}}(\mathbf{u}) = \frac{1}{2}\|\mathbf{u}\|^2 + z(\mathbf{v}, \mathbf{h}),$$

for some function $z(\mathbf{v}, \mathbf{h})$ independent of $\mathbf{u}$. Then, note that

$$\arg\min_{\mathbf{u} \in \overline{\Sigma}(\mathbf{v})} \zeta_{\mathbf{v},\mathbf{h}}(\mathbf{u}) = \arg\min_{\mathbf{u} \in \overline{\Sigma}(\mathbf{v})} z(\mathbf{v}, \mathbf{h}) + \frac{1}{2}\|\mathbf{u}\|^2 = \arg\min_{\mathbf{u} \in \overline{\Sigma}(\mathbf{v})} \frac{1}{2}\|\mathbf{u}\|^2,$$

hence the optimum does not depend on the choice of $\mathbf{h}$. Then, the claim follows from [29][Theorem 5.1]. $\qquad\square$

*Proof of lemma 6.* We will need the notion of a poset corresponding to $\mathcal{D}'$, which can be found in [1]. If we use this terminology, the sets $H_i^*$ correspond to the union of $i$ and all its ancestors in the poset. We will furthermore define $B_1$ to be the union of the ancestors and their children, $B_2$ the descendants of $i$ and $B_3$ the ideal of all elements in $B_j \setminus B_2$. First notice that the suggested solution is feasible and that $B_1, B_2$ and $B_3$ are all in $\mathcal{D}'_j$. We will now prove optimality by showing that it is optimal if we only consider the active set corresponding to $B_1, B_2$ and $B_3$. The corresponding Lagrangian is

$$\min_{\mathbf{u}} \frac{1}{2}\|\mathbf{u}\|^2 + \lambda_1(u(B_1) - 1) + \lambda_2(u(B_2) - 0) + \lambda_3(u(B_3) - 0) + \eta(\mathbf{u}^T \mathbf{1} - 1).$$

Setting the derivative wrt $u_k$ for $k \in B_1$ to zero yields the condition

$$u_k = -\lambda_1 - \lambda_2 [\![k \in B_2]\!] - \eta.$$

For any $k \in H_i^* = B_1 \setminus B_2$ we have to satisfy $-\lambda_1 - \eta = 1/|H_i^*|$, so as Ansatz take $\lambda_1 = 1/|A_1|$ and $\eta = -2/|A_1|$. Then, to satisfy the condition for any other $k \in B_1$ we can use $\lambda_2 = -\lambda_1 - \eta = 1/|H_i^*|$, which is also positive. Finally, for any $k \notin B_1$ the condition is satisfied by using the multiplier $\lambda_3 = 2/|A_i|$. $\qquad\square$

## C   Experiments

**Architectures.**   We used the following architectures in our experiments. For an exact specification of the modules please consult the documentation of `pytorch`.

```
# The network that predicts the per-pixel scores.
nn.Sequential(nn.Conv2d(3,  32, 3, padding=1),
              nn.ReLU(),
              nn.Conv2d(32, 64, 3, padding=1),
              nn.ReLU(),
              nn.Conv2d(64, 1,  3, padding=1))

# The networks that predict the log of the weights.
nn.Sequential(nn.Conv2d(3,  32, 3, padding=1),
              nn.ReLU(),
              nn.Conv2d(32, 1, 3, padding=1))
```

**Optimization.**   We have obtained the best results from the following optimizers: Adam with learning rate $10^{-3}$ for CNN+GC, and Adagrad with learning rate $10^{-2}$ for CNN. We have trained for a total of every 100 epochs, evaluated on the validation set after each epoch and chose the model with the best validation score. We have used the default parameter initialization from `PyTorch`.

# D  Two dimensional total variation code

```python
from prox_tv import tv1w_2d  # The total variation solver.
import numpy as np
import torch
from torch.autograd import Function

class TotalVariation2d(Function):
    """A two dimensional total varition function.

    Specifically, given as input ('x', 'w_r', 'w_c'), the output is

     argmin_z 0.5 |x-z|^2 + \sum_{i, j} w_r_{i,j} |x_{i, j} - x_{i, j + 1}|
                          + \sum_{i, j} w_c_{i,j} |x_{i, j} - x_{i + 1, j}|."""

    def forward(self, x, weights_row, weights_col):
        """Solves the TV problem and returns the solution.

        Arguments
        ---------
            x : [m, n] matrix holding the input signal
            weights_row : [m, n - 1] matrix holding the horizontal weights
            weights_col : [m - 1, n] matrix holding the vertical weights

        Returns
        -------
            The TV solution, matrix of size [m, n]"""
        opt = tv1w_2d(- x.numpy(), weights_col.numpy(), weights_row.numpy())
        opt = torch.Tensor(opt)
        self.save_for_backward(opt)
        return opt

    def backward(self, grad_output):
        opt, = self.saved_tensors
        grad_x = torch.zeros(opt.size())  # We always compute this derivative.

        values = np.unique(opt.numpy())  # One group for each unique value.
        groups = [(opt.numpy() == val).reshape(opt.size()) for val in values]
        for group in groups:
            grad_x.numpy()[group] = - np.mean(grad_output.numpy()[group])

        diffs_row = torch.sign(opt[:, :-1] - opt[:, 1:])
        grad_weights_row = diffs_row * (grad_x[:, :-1] - grad_x[:, 1:])

        diffs_col = torch.sign(opt[:-1, :] - opt[1:, :])
        grad_weights_col = diffs_col * (grad_x[:-1, :] - grad_x[1:, :])

        return grad_x, grad_weights_row, grad_weights_col
```

Figure 3: A pytorch function doing the forward and backward passes.