[Reviews · NeurIPS 2017]

Reviewer 1



This paper proposes a way to differentiate the process of submodular function minimization thus enabling to use these functionals as layers in neural networks. The key insight of the paper consists in the usage of the interpretation of discrete optimization of submodular functions as continuous optimization. As a concrete example the paper studies the CRF for image segmentation and creates and the graphcut layer. This layer is evaluated on the Weizmann dataset for horse segmentation and is reported to bring some improvements. I generally like the paper very much, find the description of the method clear enough. In particular, I liked the short introduction into submodular functions and their connection to min-norm-point. I have some comments that might allow to increase the impact of the paper. My comments mostly cover experimental evaluation and related works. 1. The experimental evaluation presented in Section 6 is a bit disappointing to a practitioner. The results are clearly far below state-of-the-art in image segmentation. To demonstrate the full potential of the new layer, I would recommend to plug the new layer into one of the state-of-the-art systems for image segmentation such as DeepLab (Chen et al., DeepLab: Semantic Image Segmentation with Deep Convolutional Nets, Atrous Convolution, and Fully Connected CRFs, 2016). I understand that the graphcut layer is applicable only to binary problems, but it would be very interesting to try it e.g. on one class of PASCAL VOC. 2. In particular, there is an extension of DeepLab (Chandra and Kokkinos, Fast, Exact and Multi-Scale Inference for Semantic Image Segmentation with Deep Gaussian CRFs, ECCV 2016 ) that puts a Gaussian CRF on top of the potentials learned by CNNs. They make the inference layer differentiable by connection using the fact that inference in Gaussian CRFs reduces to solving systems of linear equations. It would be very interesting to see how the graph layer compares against the Gaussian CRF layer. 3. It would make sense to mention another line of works that incorporate discrete optimization into networks. In particular, if the final loss can be computed directly from the results of discrete optimization (for example, when the whole system is trained with a structured SVM objective: Jaderberg et al., Deep structured output learning for unconstrained text recognition, ICLR 2015; Vu et al., Context-aware CNNs for person head detection, ICCV 2015). Comparison to this approach can also strengthen the paper. 4. As mentioned in line 65 a popular way of embedding algorithms into neural networks consists in unrolling a fixed number of steps of iterative algorithms into layers of neural network. This paper uses one of such iterative algorithms (a total variation solver) to do inference, so it would make sense to simply backprop through several iterations of it. Again, comparison to this approach would strengthen the paper. 5. Are there any other examples (more rich than graphcuts) of the cases when minimization of submodular functions can be plugged into neural networks? Even naming such cases together with the appropriate submodular solvers would be very valuable. 6. In terms of the potentials learned by the graphcut layer, it would be very interesting to visualize what the network has learned and, e.g., compare those with the standard potentials based on the gradient of the image. Minor comments: - Line 130 says "the Lovasz extension is linear in O", but O is a set and it is clear what the phrase means. - Line 135. [27] looks like the wrong reference - Line 143. The definition of optimal partition is never given, so it remains unclear what it is. - Line 295 says that only a small fraction of labelled pixels was used for training. It is not clear why this is done.

Reviewer 2



In this paper, the authors discuss the implement of discrete optimization on modern machine learning models. They provide an easily computable approximation to the Jacobian complemented with a complete theoretical analysis. Overall, this is a good paper and followings are detail comments. 1 In abstract, the authors said they can continuously relax the output without sacrificing guarantees. What is the original guarantees? 2 In line 33, it is better to say A^*\in argmax than A^* = \argmax since \argmax is actually a set. 3 Do you have an efficient way to test a submodular function is separable or not? 4 In your experiments, do you have a running time comparison of adding the graph cut layer?

Reviewer 3



Summary: The paper develops a method to approximate the Jacobian of the isotonic regression and the min-norm problem. The paper gives the conditions under which the approximation is correct and shows how to compute the directional derivatives exactly in polynomial time. Overall evaluation: I am not familiar with the topic in the paper and the relevant literature to comment on the significance of the results. However, this seems to be a good theory paper. Some specific comments: 1. I think the paper should include more details on the application of the results to the log-supermodular models and the end-to-end optimization problem in Equation (3). The current manuscript only mentions them briefly in Sections 1 and 2. 2. The paper should also include some examples where Theorem 3 holds. 3. I think more experiments with other benchmark image segmentation tasks would be helpful. How is the CNN+GC compared to the current state-of-the-art on those benchmark datasets? 4. The term inside the double brackets [[ . ]] in Equation (1) is confusing to me. Is it missing the cardinality function? On the other hand, F(A) on line 99 does not need the cardinality function.